# Knowledge and Attitudes of Medical and Nursing Students in a Greek University Regarding Sexually Transmitted Diseases

**DOI:** 10.3390/ijerph21030251

**Published:** 2024-02-22

**Authors:** Maria Lagadinou, Katerina Spiliopoulou, Themistoklis Paraskevas, Despoina Gkentzi, Stelios Assimakopoulos, Paraskevi Katsakiori, Leonidia Leonidou, Markos Marangos

**Affiliations:** 1Department of Internal Medicine, University General Hospital of Patras, 26504 Patras, Achaia, Greece; themispara@hotmail.com (T.P.); sassim@upatras.gr (S.A.); lydleon@yahoo.gr (L.L.); marangos@upatras.gr (M.M.); 2Medical School of Patras, University of Patras, 26504 Patras, Achaia, Greece; gkentzid@upatras.gr; 3Karamandaneio General Hospital of Patras, 26331 Patras, Achaia, Greece; katerinaspil@hotmail.com; 4Department of Pediatrics, University General Hospital of Patras, 26504 Patras, Achaia, Greece; 5Department of Infectious Diseases, University General Hospital of Patras, 26504 Patras, Achaia, Greece; 6Health Centre of Akrata, 25006 Akrata, Achaia, Greece; vkatsak@gmail.com

**Keywords:** sexually transmitted infections, medical students, nursing students, knowledge

## Abstract

Aims and Objectives: the present study aimed to assess the knowledge and attitudes of medical and nursing students at the University of Patras, western Greece, regarding sexually transmitted infections (STIs), sexual behavior and STI prevention measures, as well as the level of future healthcare professionals’ education. Method: A descriptive, cross-sectional study was conducted. A total of 231 medical and nursing students (*n* = 106 medical, and *n* = 125 nursing) completed and returned the pre-tested study questionnaire. Results: Most participants (77.5%) were females and46.1% were in the age group of 18–21 years. Syphilis, HIV/AIDS, and Hepatitis B were regarded as STIs by 65.8% of them. Medical students could predominantly list the widely known STIs compared to nursing students (*p* = 0.004). Regarding HIV/AIDS, 72.7% of the respondents reported that it is transmitted sexually and through blood transfusion. However, medical students were better informed than nursing students (*p* = 0.001). Medical students as well as students in the final year of their studies were found to be better informed about the vaccines available to prevent STIs. Regarding the question about what constitutes a risky sexual behavior, 71.4% answered sexual intercourse without the use of condom and 18.6% indicated having sex with an unknown partner. Most participants (69.7%) were satisfied with the education provided by their institution and no statistically significant difference was observed between medical and nursing students. Almost all students (97.8%) agreed that the course/subject of sex education must be included in school programs. Conclusions: A comprehensive analysis of knowledge and attitudes of Greek medical and nursing students regarding STIs, prevention measures and education level was conducted. The results of the present study could assist in the development of targeted training courses that can improve healthcare professionals’ knowledge and ability to manage STIs.

## 1. Introduction

Sexually transmitted infections (STIs)are infections passed from one person to another through intimate physical contact or sexual activity. Although more than 30 different bacteria, viruses and parasites are responsible for STIs, 8 of them show the highest incidence worldwide [1]. Four of them are responsible for curable STIs, i.e., gonorrhea, syphilis, trichomoniasis and chlamydial infections, while the rest of them cause long-lasting infections such as HIV/AIDS, genital infection of herpes simplex virus (HSV), human papillomavirus (HPV) and hepatitis B (HBV) infection [2].Globally, more than one million cases of STIs are reported every day, with 499 million cases comprising gonorrhea, syphilis, chlamydia, and trichomonas [3].Unfortunately, untreated STIs can lead to severe morbidity (infertility, pelvic inflammatory disease, ectopic pregnancy and cervical cancer)or even mortality. As no cure exists for some of them, the only way to limit their spread and eventually decrease the healthcare burden is primary prevention. In addition, the presence of non-HIV STIs increases the risk of HIV transmission [3].

STIs remain a predominant health issue that affects young adults and adolescents in both high- and low-income countries. Over the last few years and despite the efforts, the incidence of syphilis, gonorrhea, HSV, HPV, HBV, bacterial vaginosis and trichomoniasis has increased in Greece [4]. In particular, in the age group of 16–24-year-olds, young individuals are considered to be at high risk of STIs owing to their risky behavior [5].Unprotected sexual intercourse, multiple partnerships, a lack or inconsistent use of condoms and drug abuse renders younger people prone to STIs [6].Healthcare providers have a crucial role in educating adolescents and young adults on STIs as well as STIs’ transmission and preventive methods. In addition, healthcare professionals should be well informed on STIs so that they do not wrongly consider themselves being at risk when taking care of patients with STIs. Without thorough and comprehensive information, the incidence and prevalence of STIs will probably continue to rise.

To our knowledge, there is no study evaluating the knowledge and attitudes regarding STIs among Greek health sciences students. The aim of the present study was to assess the knowledge and attitudes of medical and nursing students toward STIs, sexual behavior and measures of STI prevention, as well as to evaluate the education level of these future healthcare professionals.

## 2. Methods

### 2.1. Study Design

A descriptive cross-sectional study was conducted in the University of Patras, Greece, from February to June 2023. Except for demographic characteristics, parameters like general knowledge regarding STIs, transmission and prevention methods, as well as vaccination development were evaluated. The sources that medical and nursing students use to inform themselves about STIs were also recorded.

### 2.2. Participants

Students of the departments of Medicine and Nursing were eligible to participate in the survey irrespective of their year of study. The participants were randomly selected without any exclusion criteria. All potential participants were informed in detail about the study aims as well as the data confidentiality. Students who agreed to take part in the study were asked to complete the anonymous questionnaire.

### 2.3. Instruments

Our survey was performed using a questionnaire developed by our research team based on previous studies [1,4,5,6]. The questionnaire was piloted in a small number of students prior to the beginning of the study to make appropriate adjustments. The questionnaire was written in Greek and consisted of three parts. Part A included seven questions regarding socio-demographic characteristics (age, gender, department, year of study, parenthood, and residence). Part B integrated11 questions regarding knowledge and attitudes toward STIs (the most common STIs and transmission routes), whereas Part C included 6questions about knowledge and attitudes toward STI prevention.

### 2.4. Ethical Approval

This study was approved by the Institutional Ethics Committee and was conducted according to the Guidelines of the Declaration of Helsinki. Anonymity was maintained throughout the study (Ethics Committee name: University of Patras; approval code: 4013; approval date: 7 July 2023). An overview of the study objectives was provided to potential participants, verbal consent was obtained, and subsequently, self-completion of the questionnaire took place.

### 2.5. Statistical Analysis

All fully completed questionnaires were coded and an excel workbook was then created for data entry. Statistical analysis was performed using IBM SPSS Statistics, version 28. Descriptive statistics were used to summarize all data on demographic characteristics as well as knowledge and attitudes toward STIs. Frequency distribution was worked out for proportions. Chi-square was used to determine any statistically significant association between students’ department or year of study and knowledge and attitude regarding STIs and their prevention. The level of significance was set to 0.05.

## 3. Results

### 3.1. Socio-Demographic Characteristics

A total of 300 questionnaires were randomly distributed among medical and nursing students. A total of 231 (77%) of the students (22.5% males) fully completed and returned the questionnaires and were thus enrolled in the study. Most of the participants (46.1%) were 18–21 years old, 37% were 22–25 years old and 13% were 25–30 years old. Only 3.5%of the respondents were over 30 years old. The socio-demographic characteristics of the studied population are presented in Table 1.

### 3.2. Students’ General Knowledge and Attitude toward STIs

Syphilis, HIV/AIDS and HBV infections were reported as STIs by the majority of the participants (65.8%). A statistically significant correlation was noticed between the department of study and the ability to mention STIs. More specifically, medical students were more able to list the widely known STIs compared to nursing students (AIDS, syphilis and Hepatitis B) (*p* = 0.004). Interestingly, students’ general knowledge regarding STIs (i.e., ability to list the widely known diseases and ways of transmission) was not associated with the year of study (*p* = 0.389). This indicates that the earlier years of study were not associated with less knowledge about STIs.

Most of the students (65.8%) mentioned unprotected sex as the main route of STI transmission. A statistically significant correlation between knowledge regarding transmission routes and year of study (first year of studies vs. last year of studies)and department was observed (*p* = 0.037 and *p* = 0.002, respectively). Medical students and those who were in their last year of studies in both departments gave the most correct answers regarding the transmission routes.

As far as HIV/AIDS is concerned, students were satisfactorily informed. About 72.5% of the respondents stated that HIV is transmitted sexually and through blood transfusion. Medical students were better informed than nursing ones (*p* = 0.001). However, knowledge was not statistically significantly correlated with the year of study (*p* = 0.912). The most frequent wrong answer given by the respondents was that HIV is transmitted through toilet use (3.4%).

The majority (87.5%) of respondents were aware of the prevention of HPV infection through vaccination, which is in complete agreement with the fact that 90% of respondents knew that persistent HPV infection is associated with cervical cancer. Knowledge regarding STI prevention through vaccination was statistically significantly correlated with the year of study (*p* = 0.042, Pearson: 0.113). Although all students at the end of their studies were found to be better informed about the vaccines available to prevent STIs, medical students were assessed to be better informed (*p* = 0.011, Pearson: 0.179).

### 3.3. Students’ Attitudes toward Sexual Behavior and STI Prevention

In Table 2, the primary sources that participants use to inform themselves about education issues are presented. As shown in Figure 1, family and teachers were the most frequent sources of information about STIs. Moreover, the media was mentioned as an important source of information for medical students (29.2%). When participants were asked what they considered risky sexual behavior, 71.4% answered sexual intercourse without the use of a condom and 18.6% mentioned having sex with an unknown partner without using a condom. It is noteworthy that almost all students (92.2%) knew that there are several STIs transmitted with oral sex. Figure 2 shows the answers that participants gave regarding risky sexual behavior.

A total of 68% of the participants considered a condom an effective method of preventing STIs, whereas 11.3% answered that sexual abstinence is an effective method for STI prevention as well. Knowledge regarding STI prevention was not statistically significantly correlated with the year of study independently of the department (*p* = 0.103, Pearson: 0.161). Table 3 and Figure 3 show the frequency of condom use among medical and nursing students. No statistically significant correlation was observed between the year of study (*p* = 0.431, Pearson: 0.158) or the study department (*p* = 0.428, Pearson: −0.009) and condom use frequency. When participants were asked about the factors affecting their sexual behavior (having sex with an unknown or multiple partners, without using a condom), both medical and nursing students answered that they were mostly affected from their personality (32.9%), secondly from their family (19%) and thirdly from their social environment (16%).

### 3.4. Students’ Education on STIs

Most of the students (69.7%) were satisfied with the education provided by their institution on STIs (Table 4). No statistically significant difference was observed between medical and nursing students (Figure 4). Most of the students agreed that a sex education course must be included in school programs.

## 4. Discussion

STIs are responsible for multiple health, social and economic problems worldwide. Additionally, inadequate knowledge and wrong attitudes toward STI transmission routes and prevention measures, especially among healthcare professionals, can be devastating as they are the ones to care for and inform patients. According to the results of the present study, students in the faculties of Medicine and Nursing at the University of Patras, western Greece, had a satisfactory level of knowledge regarding the most widely known STIs. However, medical students seem to be better informed. In a similar study by Orisatoki RO et al., the knowledge of medical students about STIs was found to be high [7].Subbarao NT et al. published a similar study on college students and reported that most of them had heard about STIs and that STIs other than HIV were known by only around 64% of students who participated in the survey [2]. An important finding of the present study was that most medical and nursing students had enough knowledge regarding HIV/AIDS (STIs, transmission routes, etc.). More recently, Raia-Barjat et al. reported poor knowledge of STIs and STI prevention measures as well as risky behaviors in French health students and highlighted the necessity of good training [8]. As reported in previous studies conducted with university students in Malaysia [9] and Turkey [10], HIV/AIDS remains one of the most widely known STIs worldwide. In the cross-sectional study by Shireraw Y. et al., high awareness of HIV/AIDS and its mode of transmission was observed in preparatory students [6]. This is an encouraging finding which should be further strengthened by organizing AIDS awareness campaigns among the young population.

Hopefully, almost all medical and nursing students that participated in the study were aware of the association between HPV infection and cervical cancer, as well as the prevention of HPV infection through vaccination. Similarly, Singh et al. reported that the majority of medical students (57.57%) knew about the availability of the HPV vaccine through sources like the Internet, magazines and others [11]. Khatiwada M. et al. reported that 97.2% of medical, nursing and social sciences students who participated in their study have heard of the association between HPV infection and cervical cancer [12]. In the same study, over 90% of the participants claimed that HPV is the leading cause of cervical cancer, indicating that students were well educated about HPV. In contrast to our study, 66% of the participants had heard of the available HPV vaccines [11]. Aksoy N et al. reported that the knowledge of health students regarding HPV, HPV vaccination and cervical cancer was rather high. However, knowledge gaps exist and need to be corrected through educational programs [13].

Most of the participants in the current study were informed about STIs through family, teachers and mass media. In a similar study conducted by Mahmood et al., teachers, the Internet and the media were reported as the main sources of information for medical students [1]. These findings emphasize the need to improve the role of teachers in STI awareness programs as information provided by unqualified personnel may be incomplete or misleading. Subbarao and Akhilesh also reported that students were informed about these infections through teachers, the Internet and newspapers/magazines [2,6]. Teachers and mass media were also mentioned as the main information sources on STIs among secondary school students [12]. The findings of the present study are in line with the aforementioned studies. Family and teachers were the most frequent sources of information reported by the respondents. Moreover, the media was mentioned as an important source of information for medical students (29.2%).

Regarding STI prevention measures, it is important that most participants replied that condom use protects against STI transmission. In accordance with our findings, poor use of condoms and a high number of unknown partners have also been reported as the main factors responsible for the high incidence of STIs in an Italian survey comprising 294 subjects with STIs [14,15]. It is hopeful that almost all participants sometimes use a condom. However, a condom promotion program could be effective in preventing high-risk sexual behaviors among urban young adults, as shown in the study by Kennedy et al. [16].

Although most students were satisfied with the education provided by their institution, with no statistically significant difference between medical and nursing students, a more structured and thorough education on STIs is mandatory. Unfortunately, sexual education programs in Greek schools are not currently available, something that is deemed necessary to change by all study participants (97.8%). Intervention programs providing sex education to younger ages, even in secondary school, have been reported to result in a marked improvement in the knowledge of students about STIs. Additionally, they have been associated with a positive change in their attitude toward them, especially considering that not all secondary students are sexually active [6].

To our knowledge, this is the first study in Greece evaluating knowledge and attitudes regarding STIs in health sciences students (medicine and nursing). The response rate of participants was 77%, which was satisfactory. However, the present study shows certain limitations that must be addressed. First, it is a single-center study with a small number of participants. Since the study was conducted among future health professionals, conclusions cannot be generalized to the Greek population. Secondly, participation in the study was voluntary; therefore, students feeling uncomfortable about STIs may have deliberately decided not to participate, causing selection bias. In addition, although participants were reassured about the anonymity of their participation in the study in advance of completing the questionnaire, their honesty cannot be evaluated. However, our descriptive, cross-sectional study provides useful data on what trends can be expected in the students’ knowledge and attitudes toward STIs, the way in which they spread and the available preventive measures. Finally, the need for structured educational programs is highlighted based on both the students’ knowledge and attitudes toward STIs.

## 5. Conclusions

A comprehensive analysis of the knowledge and attitudes of medical and nursing students regarding STIs, prevention measures and education level was carried out in the University of Patras, western Greece. Most of the participants were aware of the main STIs and the routes of transmission. A significantly high percentage of the participants knew that condom use protects against STIs. Although school education and mass media were reported as the main sources of information for health students, family still plays a crucial role in their education. The results of our study highlight the urgent need for well-designed, continuous sexual education programs to increase students’ awareness about STIs and their prevention.

## Figures and Tables

**Figure 1 ijerph-21-00251-f001:**
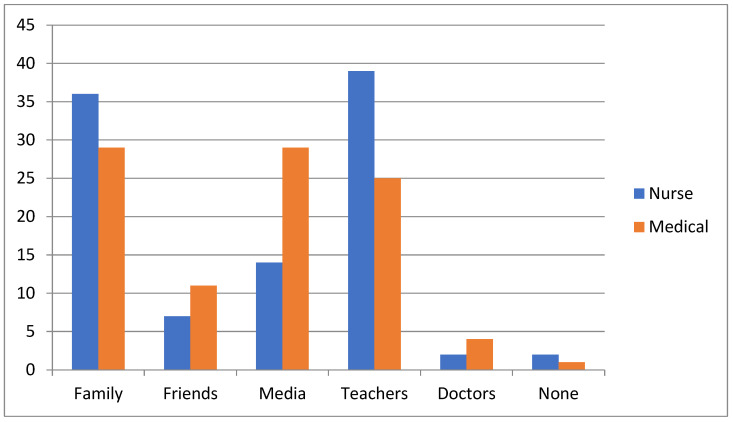
Sources that medical and nursing students use to inform themselves about STIs.

**Figure 2 ijerph-21-00251-f002:**
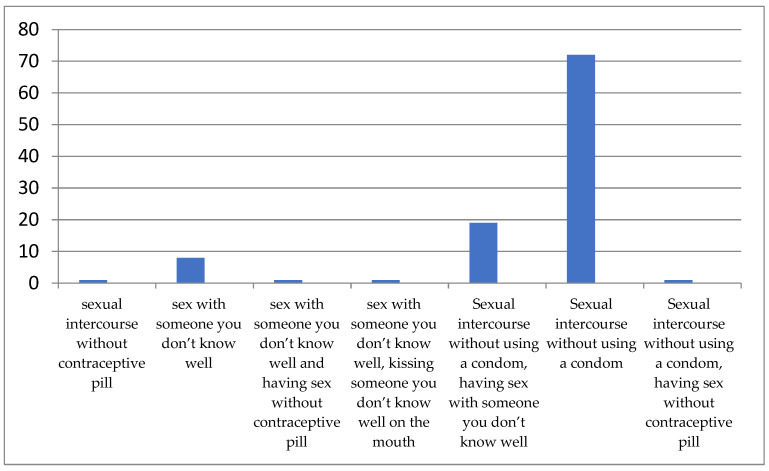
Risky sexual behavior (frequency refers to the absolute number of respondents who gave the specific answer).

**Figure 3 ijerph-21-00251-f003:**
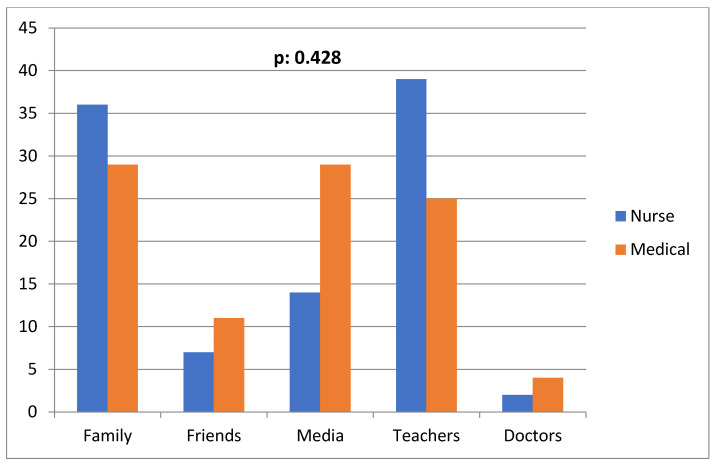
Frequency of condom use in medical and nursing students (*p* value refers to the statistical difference inthe overall frequency of condom use between medical and nursing students).

**Figure 4 ijerph-21-00251-f004:**
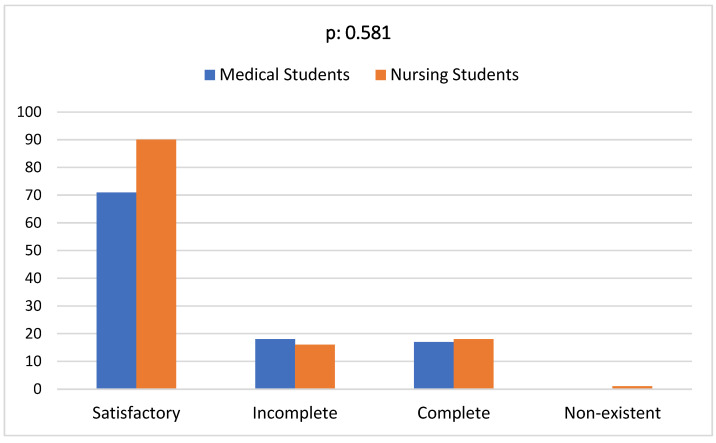
Satisfaction of medical and nursing students regarding the education level of STIs. *X* axis represents the level of education. *Y* axis represents the frequency of each level. No statistically significant differences were noted between medical and nursing students (*p* = 0.581).

**Table 1 ijerph-21-00251-t001:** Socio-demographic characteristics of the study population.

Socio-Demographic Characteristics	Number (Percentage)	
	Medicine	Nursing
Total number of participants	106 (45.9%)	125 (54.1%)
Gender		
Male	30 (28.3%)	22 (17.6%)
Female	76 (71.7%)	103 (82.4%)
Age (years)		
18–21	21 (19.8%)	85 (68.0%)
22–25	55 (51.9%)	31 (24.8%)
25–30	27 (25.5%)	4 (2.4%)
>30	3 (2.8%)	5 (4.0%)
Ethnicity		
Greek	102 (96.2%)	119 (95.2%)
European	4 (3.8%)	4 (3.2%)
Other	0	2 (1.6%)
Year of study		
First	9 (8.5%)	30 (24%)
Last	27 (25.5%)	27 (21.6%)
Other	68 (65.1%)	68 (54.4%)
Type of Family		
Two parents	101 (95.3%)	106 (84.8%)
Divorced	3 (2.8%)	15 (12.1%)
Fatherless	2(1.9%)	4 (3.2%)

**Table 2 ijerph-21-00251-t002:** Primary source used to inform themselves about education issues.

		Department		Total	*p* Value
		Medical Students	Nursing Students		
Your first source of information regarding sexual issues	Family	32 (30.2%)	45 (36%)	77	1.000
	Friends	12 (11.4%)	9 (7.2%)	21	1.000
	Media	31 (29.2%)	17 (13.6%)	48	0.022
	Teachers	26 (24.5%)	49 (39.2%)	75	1.000
	Doctors	4 (3.8%)	3 (2.4%)	7	1.000
	None	1 (0.9%)	2 (1.6%)	3	1.000
Total		106 (100%)	125 (100%)	231	

**Table 3 ijerph-21-00251-t003:** Frequency of condom use.

		Department		Total	*p* Value
		Medical Students	Nursing Students		
Condom use	Sometimes	84 (83.2%)	105 (88.2%)	189	1.000
	When necessary	7 (6.9%)	5 (4.2%)	12	1.000
	Never	6 (5.9%)	8 (6.7%)	14	1.000
	Always	2 (1.9)	0	2	1.000
	Often	2 (1.9)	1 (0.9%)	3	1.000
Total		101 (100%)	119 (100%)	220	

**Table 4 ijerph-21-00251-t004:** Students’ satisfaction regarding the education they have already received.

Question: Your Education about Sexually Transmitted Infections So Far Is:			
		Frequency	Percent
Valid	Satisfactory	161	69.7%
	Incomplete	34	14.7%
	Complete	35	15.2%
	Non-existent	1	0.4%
	Total	231	100

## Data Availability

The data that support the findings of this study are available from the corresponding author upon reasonable request.

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
