# Peer review of "Knowledge and Attitudes of Medical and Nursing Students in a Greek University Regarding Sexually Transmitted Diseases"

_ijerph, 2024, doi:10.3390/ijerph21030251_

Round 1

Reviewer 1 Report

Comments and Suggestions for Authors

The paper presents an important topic - that is, risky sexual behaviors among college students in Greece. The study design, a descriptive cross-sectional study, is not novel during this decade of risky sexual behaviors. However, there are two areas to address. They include - (1) the statistical presentations could include correlations and (2) the paper could benefit from editing for the reading public to better understand the key contexts of the study. 

Comments on the Quality of English Language

Moderate editing is recommended.

Author Response

Dear Reviewer thank you for your comments!

Reviewer 2 Report

Comments and Suggestions for Authors

Estimated Authors,

I've read with great interest your paper on the Knowledge and Attitudes of Medical and Nursing Students in a Greek University regarding Sexually Transmitted Diseases. Your study has stressed that, despite a background healthcare formation of study participants, some knowledge gaps still remain, leaving significant space for future interventions aimed to improve the overall understanding of this topic and preventive measures.

Despite its potential significance and interest, the present paper still requires some improvements, according to the present reviewer. More precisely:

1.  Have you performed a preventive power analysis? Was a minimum sample size calculated? Please discuss whether the sample size could be considered representative of the targeted population or not, and how the population of university-level students could be perceived as representative or not of young-adults from Greece.

2. Figure 1 + 2 could be merged and should be simplified from 3D to 2D in order to guarantee a better perception of reported values and differences.

3. Figure 3 was unclear as improperly paginated, please fix.

4. Table 2, 4, 5 could be merged.

5. What about the internal consistency of the questionnaire? Have you performed a preventive analysis (e.g. calculation of Crombach's alpha

6. When dealing with correlation tests, please provide the point value of the correlation coefficient, and not only that of p value.

7. Some typos are scattered across the main text; please fix it.

Author Response

(The authors gave the same response as above.)

Reviewer 3 Report

Comments and Suggestions for Authors

Review the wording of the summary, it is mentioned 2 times to nursing students when it should be medicine and nursing.

The discussion must compare its results with other similar studies and establish a true academic discussion, explaining the results obtained and whether or not they are similar to other studies.

The presentation of results can be improved with graphs in some cases.

Table 1 reports values ​​of the P value, however the sociodemographic characteristics of the population are not part of the study. Just as the P value in the relationship of these variables with the race is not understood.

Comments on the Quality of English Language

It is necessary to improve English writing to facilitate better understanding of readers.

Author Response

Dear Reviewer.

Thank you for your comments!

Reviewer 4 Report

Comments and Suggestions for Authors

Dear authors,

It was a pleasure for me to review this manuscript dealing with Greek health science students' knowledge and attitudes towards sexually transmitted infections.

It is very important that health professionals have good training in this matter so that they can correctly advise their patients. For this reason I find it very interesting that the authors try to evaluate the knowledge and attitudes of these professionals.

With the sole objective of improving the quality of the manuscript, I will allow myself to make a series of comments:

1. In the Summary the acronym ETS is used without previously describing it (line 14)

2. I suggest changing the term STD to STI. The change in concept, coined by the World Health Organization (WHO) in 1998, which replaces the terminology of Sexually Transmitted Diseases (STD) with Sexually Transmitted Infections (STI) is based on the fact that the term "Disease" is inappropriate to designate those infections that are asymptomatic and that go unnoticed by people with sometimes irreversible consequences.

3. Section 2.1 covers too many aspects. I suggest there be a section 2.1. Study design; A section 2.2, Participants; A section 2.3, Instruments; 1 section 2.4, Ethical considerations.

4. In the study design section, the variables that were considered in the study should be described.

5. For participants, it is necessary to define the sample size in addition to the inclusion and exclusion criteria.

6. In the instrument section, it must be explained whether a previously validated tool has been used and, if not, explain the validation data of this study.

7. The statistical study seems very poor to me. Only a bivariate study with qualitative variables was carried out. No quantitative variable was used to relate the analysis? Is there no multivariate model to adjust for confounding factors?

8. The conclusions are very simple and do not faithfully support the objective of the study.

Thanks

Kind regards

Author Response

Dear reviewer,

thank you for your valuable comments

Round 2

Reviewer 2 Report

Comments and Suggestions for Authors

Estimated Authors,

Your study has been improved according to my recommendations.

As a consequence, I've no further requests.

By the way, please note that Figure 4 remains quite difficult to appreciate because of pagination errors: I leave to the assistant editors of MDPI the management of this issue.

Author Response

Thank you again for your valuable comments

Reviewer 4 Report

Comments and Suggestions for Authors

Dear authors,

It was a pleasure for me to review this second improved version of this manuscript that attempts to analyze the knowledge and attitudes of medical and nursing students at a Greek university regarding sexually transmitted diseases.

It could be seen that in this second version the quality of the manuscript was improved and the authors followed the recommendations provided by the different reviewers.

I suggest you check the reference list. There are many words that lack space between them.

To conclude, I would like to say that this second version of the manuscript was much more interesting than the previous one.

Kind regards

Author Response

(The authors gave the same response as above.)
